# Prediction Models of Primary Membranous Nephropathy: A Systematic Review and Meta-Analysis

**DOI:** 10.3390/jcm12020559

**Published:** 2023-01-10

**Authors:** Chanyu Geng, Liming Huang, Yi Li, Amanda Ying Wang, Guisen Li, Yunlin Feng

**Affiliations:** 1Department of Nephrology, Sichuan Provincial People’s Hospital, University of Electronic Science and Technology of China, Chengdu 610072, China; 2Chinese Academy of Sciences Sichuan Translational Medicine Research Hospital, Chengdu 610072, China; 3The Faculty of Medicine and Health Sciences, Macquarie University, Sydney 2109, Australia; 4The Renal and Metabolic Division, George Institute for Global Health, UNSW, Sydney 2042, Australia; 5Department of Renal Medicine, Concord Clinical School, The University of Sydney, Sydney 2006, Australia

**Keywords:** prediction models, primary membranous nephropathy, systematic review, meta-analysis

## Abstract

Background: Several statistical models for predicting prognosis of primary membranous nephropathy (PMN) have been proposed, most of which have not been as widely accepted in clinical practice. Methods: A systematic search was performed in MEDLINE and EMBASE. English studies that developed any prediction models including two or more than two predictive variables were eligible for inclusion. The study population was limited to adult patients with pathologically confirmed PMN. The outcomes in eligible studies should be events relevant to prognosis of PMN, either disease progression or response profile after treatments. The risk of bias was assessed according to the PROBAST. Results: In all, eight studies with 1237 patients were included. The pooled AUC value of the seven studies with renal function deterioration and/or ESRD as the predicted outcomes was 0.88 (95% CI: 0.85 to 0.90; I^2^ = 77%, *p* = 0.006). The paired forest plots for sensitivity and specificity with corresponding 95% CIs for each of these seven studies indicated the combined sensitivity and specificity were 0.76 (95% CI: 0.64 to 0.85) and 0.84 (95% CI: 0.80 to 0.88), respectively. All seven studies included in the meta-analysis were assessed as high risk of bias according to the PROBAST tool. Conclusions: The reported discrimination ability of included models was good; however, the insufficient calibration assessment and lack of validation studies precluded drawing a definitive conclusion on the performance of these prediction models. High-grade evidence from well-designed studies is needed in this field.

## 1. Introduction

Primary membranous nephropathy (PMN) is a glomerulonephropathy that affects all ethnicities, all regions, and all ages [1]. It has relatively “benign” presentations and was used to be governed for a long time by the rule of thirds, with a third of patients responding to treatments to variable extents, a third progressing to renal insufficiency, and a third undergoing spontaneous remission [2]. Despite of substantial advances in research on underlying mechanisms of PMN in the past twenty decades [3], the treatment remains controversial [1]. The KDIGO 2021 guideline recommends treating PMN according to a risk classification which includes four risk categories [4]. Strategies that can enable clinicians to identify patients who will benefit from treatments would be useful for providing individualized precision therapies while avoiding unnecessary adverse effects.

In response to these unmet needs, several statistical models for predicting prognosis of PMN have been proposed. The Toronto risk score, firstly proposed in 1992, used kidney function and proteinuria variables to predicted the risk of renal failure [5]. Some other prediction models using renal function deterioration as their predicted outcomes have also been proposed [6,7,8,9]; however, most of these prediction models have not been accepted as widely as the international risk-prediction tool in IgA nephropathy [10] due to the lack of clinical validation. In addition, the published prediction models mainly focused on forecasting renal function deterioration, giving insufficient attention to the disease remission profile after treatments, which is also essential for the management of PMN.

Therefore, we conducted this systematic review and meta-analysis to summarize current prediction models for PMN, aiming to understanding the gap between ongoing studies and clinical needs and providing clues for future investigations in this field.

## 2. Materials and Methods

### 2.1. Data Sources and Searches

A systematic search according to the Preferred Reporting Items for Systematic Review and Meta-Analyses statement [11] was performed for eligible studies published up to 19 September 2022 in MEDLINE via PubMed (from 1946 through September, 2022) and EMBASE (from 1980 through September, 2022). The search terms included text words and medical subject headings relevant to prediction models and primary membranous nephropathy (see Appendix A). This study was registered on PROSPERO (Identifier# CRD42022363539).

### 2.2. Study Selection

English studies that developed any prediction models including two or more predictive variables were eligible for inclusion. Prediction models might be presented in different forms, including risk scores, equations, on-line calculators, etc. The study population was limited to adult patients with pathologically confirmed PMN. No constraint was imposed on modeling algorithms or publication years.

Two reviewers (G.C.Y. and H.L.M.) independently conducted the review process. Titles and abstracts of all returned records were carefully reviewed. Duplicates, non-original studies (e.g., reviews, editorial commentaries, and correspondence), studies that investigated risk factors instead of prediction models for poor prognosis of PMN, and studies irrelevant to PMN were excluded. Abstracts with sufficient information reported were considered eligible. Reference lists from full-text reviewed publications were also manually scanned to identify any relevant studies. Any discrepancy was adjudicated by a third reviewer (F.Y.L.).

### 2.3. Outcomes

The outcomes in eligible studies should be events relevant to the prognosis of PMN, either disease progression or remission profile after treatments.

### 2.4. Data Extraction and Quality Assessment

Two reviewers (G.C.Y. and H.L.M.) independently extracted data from included studies using a standardized sheet. Disagreements were resolved by the third reviewer (F.Y.L.). Data extracted included authors, publication year, geographical origin, study population, number of patients, age and composition of study population, follow up duration, predicted outcomes, number of predictive variables, statistical modeling approaches, discrimination performance, calibration performance, and report of internal validation and external validation. The discrimination indices included C-statistics, specificity (SPEC), sensitivity (SEN), positive likelihood ratio (PLR), positive predictive value (PPV), negative likelihood ratio (NLR), and negative predictive value (NPV). The calibration indices included results of Hosmer–Lemeshow test and calibration plot.

### 2.5. Critical Appraisal of Included Studies

Two reviewers (G.C.Y. and H.L.M.) independently assessed the risk of bias of the included studies based on the Prediction model Risk Of Bias ASsessment Tool (PROBAST) [12] which contains four domains and twenty signaling questions. The overall risk of bias of an individual study was assessed as low only if all four domains were rated as having low risk of bias.

### 2.6. Data Synthesis and Analysis

The Review Manager (RevMan) (version 5.2, The Cochrane Collaboration, London, UK) and STATA 14.0 (Stata Corporation, College Station, TX, USA) software were used for data synthesis. All analysis procedures were performed through a double-checked process by two reviewers (F.Y.L. and G.C.Y.) to avoid data entry errors. A Nightingale rose chart was generated to illustrate the composition of predictive variables using Microsoft Excel. Discrimination assessment results for studies that had only provided ROC curves without detailed values were extracted from the ROC curves using GetData Graph Digitizer (version 2.2.6) [13]. A summary ROC (sROC) curve with a 95% confidence interval (CI) was generated using the hierarchical summary receiver operating characteristics (HSROC) model [14] to evaluate the pooled discrimination ability of published prediction models for PMN prognosis. Statistical heterogeneity was estimated using I^2^ statistic [15], assessed as low if I^2^ of <25%, moderate if I^2^ ranged from 26% to 75%, and high if I^2^ of >75%. The statistical significance was set at *p* < 0.05. Pooled results of sensitivity and specificity with 95% CIs were also calculated from the HSROC model. Intraclass correlation coefficient (ICC) was used to assess interstudy variations in sensitivity and specificity. A Fagan diagram was used to test the posttest probability [16]. Sensitivity analyses and funnel plots analysis for publication bias were not applicable due to the limited number of studies included in the meta-analysis.

## 3. Results

### 3.1. Search Findings

A total of 1684 records were identified from literature searching after removing duplications. Thereafter, 33 citations were kept for full text review after title and abstract screening, among which 10 citations were further excluded due to unavailability of full-text article. Fifteen articles were excluded for having not reported prediction models or including insufficient data for the meta-analysis, leaving eight studies finally included in this systematic review (see Figure 1).

### 3.2. Study Characteristics

Among these eight studies involving 1237 patients, six studies reported prognostic prediction models on renal function deterioration [5,6,7,8,9,17], while the other two reported prediction models on proteinuria remission [18,19]. Five and three of the eight studies were retrospective and prospective, respectively. The number of patients in each individual study population substantially varied from 57 to 439. The percentages of male patients in individual studies ranged from 44.9% up to 66.7%. The shortest follow-up period was six months. Basic characteristics of included studies were shown in Table 1.

The number of predictive variables in these eight models varied from two to six. The pooled frequencies of reported predictive variables indicated the four mostly reported predictive variables were, in the descending order, proteinuria, serum anti-phospholipase A_2_ receptor (PLA_2_R) antibody, renal function, and age (Figure 2). All other predictive variables had been reported only once. It should be noted only two of the eight studies had reported calibration performance assessments of their prediction models. The lack of internal and external validation was also prominent. A summary of prediction performance assessments was shown in Table 2.

### 3.3. Pooled Discrimination Ability

Studies using proteinuria remission as their predicted outcomes did not allow a meta-analysis due to the limited number. The pooled AUC value of the six studies with renal function deterioration and/or ESRD as the predicted outcomes was 0.88 (95% CI: 0.85 to 0.90; I^2^ = 77%, *p* = 0.006) (Figure 3). The paired forest plots for sensitivity and specificity with corresponding 95% CIs for each of these six studies indicated the combined sensitivity and specificity were 0.76 (95% CI: 0.64 to 0.85) and 0.84 (95% CI: 0.80 to 0.88), respectively (Figure 4). ICCs (95% CI) assessing interstudy variations in sensitivity and specificity were 0.12 (0.00 to 0.30) and 0.03 (0.00 to 0.08), respectively. Assuming a 20% prevalence of renal function deterioration in PMN, the Fagan nomogram showed that the posterior probability of renal function deterioration would be 55% if the predicted outcome was positive, and the posterior probability of the absence of renal function deterioration would be 7% if the predicted outcome was negative (Appendix A).

### 3.4. Critical Appraisal

All six studies included in the meta-analysis were assessed as high risk of bias according to the PROBAST tool (Figure 5). The analysis domain was mostly rated as having high risk of bias due to having an event per variable (EPV) ratio of <10, including predictive variables following a “first uni- then multi-” variable regression procedure, or lacking internal validation.

## 4. Discussion

The findings of this systematic review and meta-analysis indicated the published prediction models for PMN were relatively few in number compared with those for other kidney diseases such as acute kidney injury. The pooled discrimination ability of included prediction models was good; however, the insufficient calibration assessments and lack of validation studies precluded drawing a definitive conclusion on the performance of these prediction models. In addition, all included studies suffered from high risk of bias.

Among all predictive variables reported in the included studies, proteinuria and renal function variables were the two most frequently used. This is consistent with literature that indicated serum creatinine and proteinuria were the oldest predictors for risk of progressive kidney disease [1]. The majority of other predictive variables in this systematic review were laboratory findings from serum or urine samples. Serum anti-PLA2R antibody level was the third most frequently used predictive variable, even though it was still considered as a non-validated yet clinical useful predictor [1]. A few included predictive variables are not widely utilized in clinical practice, such as urinary α1 microglobulin corrected by creatinine and neutrophil-to-lymphocyte ratio. It is worth noting that none of these prediction models included variables relevant to therapeutic regimens. Recently, therapy protocols with CD-20 monoclonal antibodies have revolutionized the guideline on the management of PMN [4,21,22,23,24]. Including variables relevant to treatments might help to improve the performance of prediction models; however, this hypothesis calls for results from well-designed studies with long-term observation.

Most published prediction models for PMN focused on forecasting so-called hard endpoints, including renal function deterioration, ESRD, and death. Only two studies in this systematic review used proteinuria remission as their predicted outcomes [18,19]. Although the natural history of PMN was traditionally considered benign for the risk of renal progression, this disease has remained one of the leading causes of renal failure among various primary glomerulopathies in the US and Europe [4]. Long-term nephrotic proteinuria not only implies the absence of remission, but also increases risk of thromboembolic events [2]. Even for patients with proteinuria of less than 6 g/day, long-term nephrotic proteinuria may cause consistent hypoalbuminemia, which in turns induce overt edema. Therefore, treatments are often required, including both non-immunosuppressive supportive treatment and immunosuppressive therapies. Prediction models targeting at disease remission may help to select patients who might benefit from treatments, customize therapeutic regimens, and avoid unnecessary adverse effects. It is for this consideration that more accurate prediction models are needed to predict the efficacy after treatments in PMN and assist clinical decision making.

Although the overall discrimination performance of included studies in this meta-analysis was good, reflected by the pooled C statistic higher than 0.85 [6,7,8,18,19], most of these models had not undergone validation. Only two studies had been internally validated [6,8]; only one study had been externally validated [5,20]. Model validation is a critical approach to confirm the robustness of a prediction model. Internal validation helps to verify the reproducibility of the modelling process and prevent overfitting of the model that might result in overestimation of the performance of the model [25], therefore, being considered mandatory by the PROBAST tool. Four prediction models in this systematic review were rated as having high risk of bias due to the lack of internal validation. External validation is not a requirement of the PROBAST tool. It is used to verify the consistency of performance of the model in different time periods, different regions, or different populations; however, predictive performance may worsen substantially on external validation [26]. It should be noted the majority of included studies in this meta-analysis did not report calibration results. The lack of calibration assessment and validation prevented us from drawing definitive conclusions on the performance of these prediction models. This might be one of the reasons why current prognosis prediction models for PMN have not been widely used in clinic yet.

To the best of our knowledge, this is the first systematic review and meta-analysis on the prognostic prediction models for PMN. There are a few limitations to be mentioned. First, the number of included studies was small, preventing us from conducting sensitivity analysis to explore the high heterogeneities or publication bias analysis. Second, partial data for generation of the sROC curve had been extracted through the GetData software, which might have led to some errors. Third, the quality of each included studies was rated as low, reflected by the high risk of bias assessment from the PROBAST tool. High-grade evidence from well-designed studies is needed in this field.

## 5. Conclusions

This systematic review and meta-analysis indicated the published prediction models for PMN were relatively few in number compared with those for other kidney diseases. The pooled discrimination ability of the included models was good; however, the insufficient calibration assessment and lack of validation studies precluded drawing a definitive conclusion on the performance of these prediction models. High-grade evidence from well-designed studies is needed in this field.

## Figures and Tables

**Figure 1 jcm-12-00559-f001:**
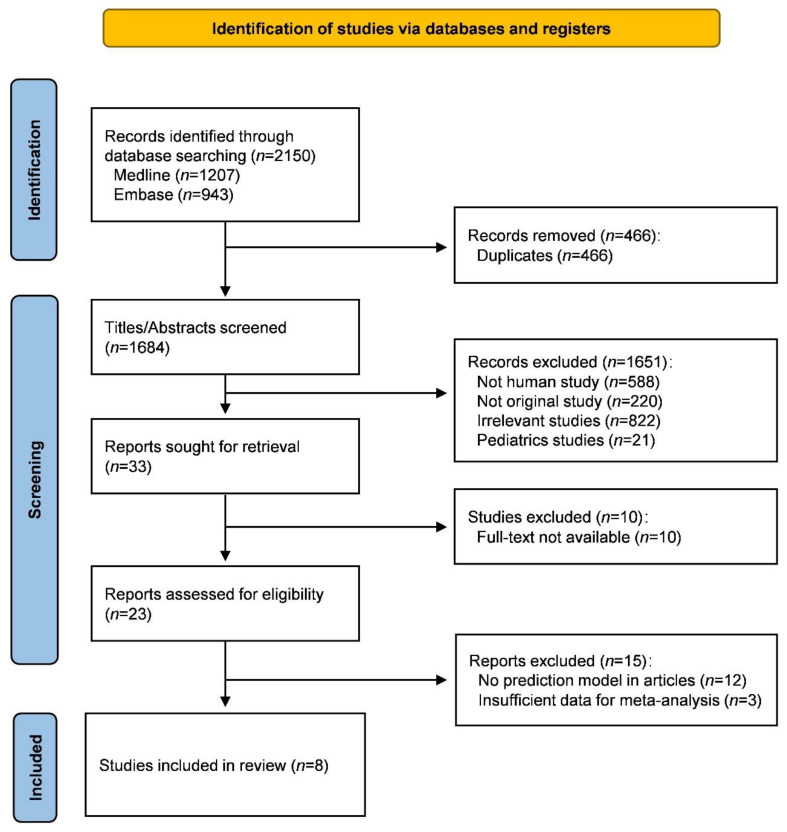
PRISMA flow chart.

**Figure 2 jcm-12-00559-f002:**
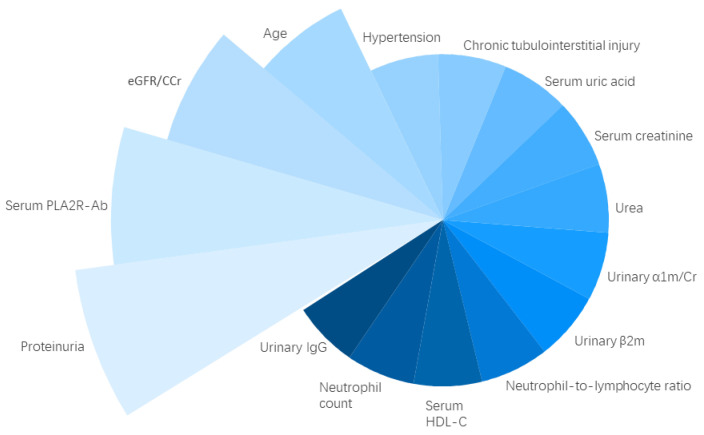
The frequencies of individual predictive variables in the included prediction models. Abbreviations: eGFR, estimated glomerular filtration rate; HDL-C, high-density lipoprotein cholesterol; PLA_2_R-Ab, PLA_2_R antibody; IgG, Immunoglobulin G; α1m/Cr, α1-microglobulin corrected by creatinine; β2m, β2-microglobulin.

**Figure 3 jcm-12-00559-f003:**
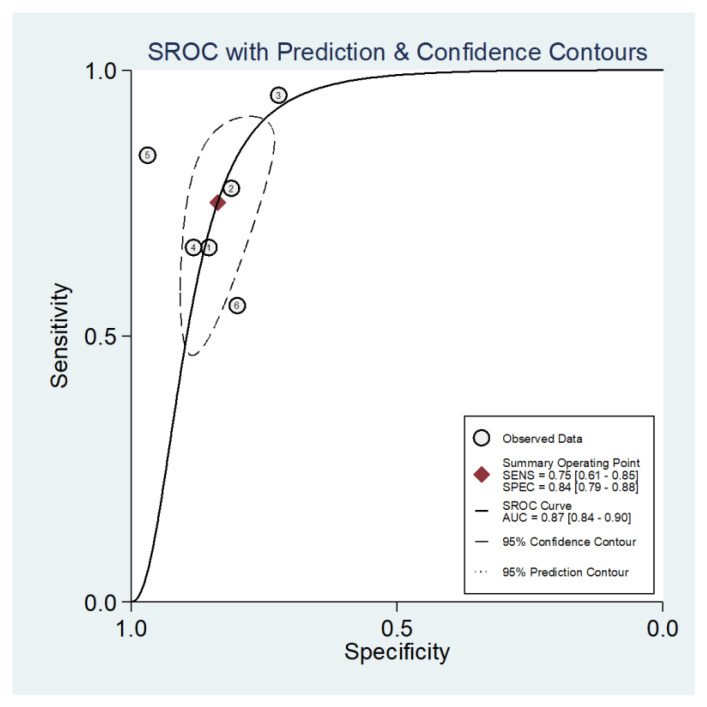
sROC curve with prediction and confidence contours of the six prediction models with renal function deterioration as predicted outcomes.

**Figure 4 jcm-12-00559-f004:**
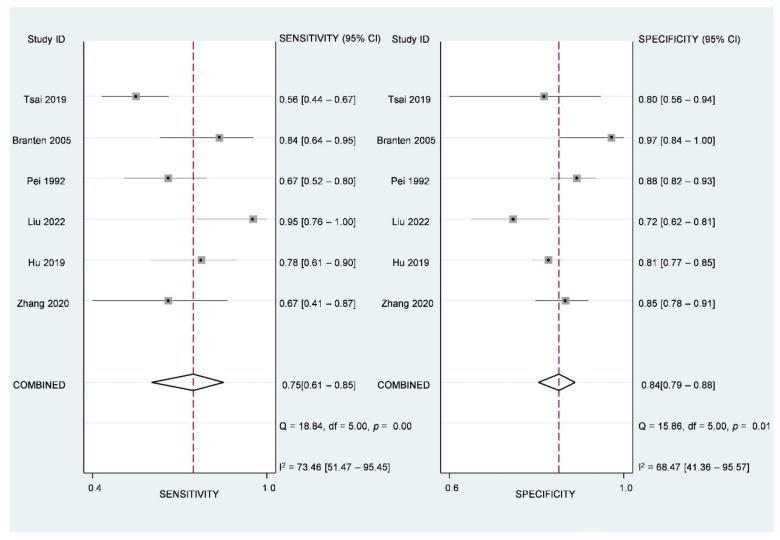
Forest plots of the pooled sensitivity and specificity with corresponding 95% CIs of the six prediction models with renal function deterioration as predicted outcomes.

**Figure 5 jcm-12-00559-f005:**
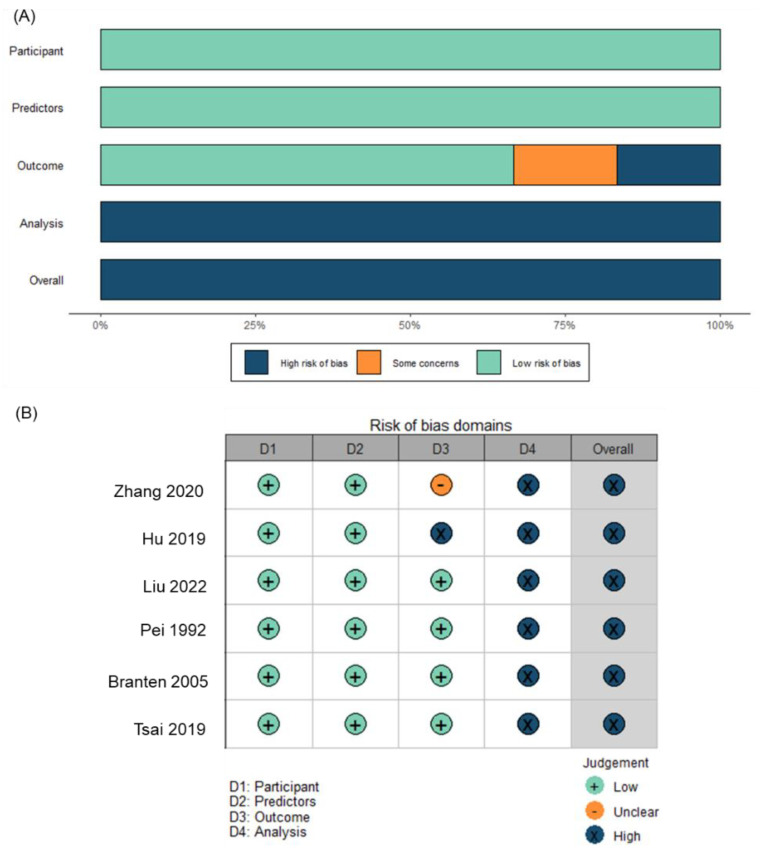
Risk of bias assessment of the six prediction models with renal function deterioration as predicted outcomes. Note: (**A**) Risk of bias summary graph; (**B**) traffic light graph for the risk of bias.

**Table 1 jcm-12-00559-t001:** Basic characteristics of included studies.

Author/Year	Region	Study Design	Study Population				Outcome
			Definition	No. (Male %)	Age (y) Mean ± SD/Median (Range)	F/u Duration (m) Mean ± SD/Median (Range)	
Zhang, 2020 [6]	China	Retrospective	Patients who received a biopsy-based diagnosis of PMN between January 2010 and December 2018	141 (57.4)	51.5 ± 14.1	30 (21,45)	Poor renal outcome defined as an eGFR decrease of ≥50% from the baseline level or progression to ESRD during f/u or death caused by MN
Hu, 2019 [7]	China	Retrospective	PMN patients diagnosed by renal biopsy from January 2009 to December 2013	439 (52.4)	56 (15,83)	38.73 ± 19.35	A combination of renal function deterioration defined as a reduction of eGFR ≥30% comparing to baseline, ESRD, or death
Liu, 2022 [8]	China	Retrospective	PMN patients who underwent native renal biopsy between January 2012 and June 2018	111 (61.3)	57 (41,66)	40 (12,92)	Renal function deterioration defined as a reduction of eGFR ≥ 20% compared with baseline or onset of ESRD
Pei, 1992 [5]	Canada	Prospective	PMN patients in Toronto Glomerulonephritis Registry from August 1974 to August 1987	184 (66)	43 ± 17	69.6 ± 48	Chronic renal insufficiency, defined as a creatinine clearance of less than 60 mL/min/1.73 m^2^ persisting for ≥12 months
Branten, 2005 [9]	The Netherlands	Prospective	PMN patients confirmed by biopsy since 1995	57 (66.7)	48 ± 16	53 ± 23	Renal death, defined as a serum creatinine exceeding 1.5 mg/dL or a rise of serum creatinine of >50%
Tsai, 2019 [17]	China	Retrospective	Patients who were >20 years old and had received a first renal biopsy for diagnoses of PMN from January 2003 to December 2013	99 (53.5)	57.75 ± 16.57	74.4 ± 38.4	Patient death and renal death (ESRD)
Wu, 2022 [18]	China	Retrospective	patients diagnosed of PMN by renal biopsy from January 2019 to February 2021	79 (64.6)	46(37,54)	12	Complete remission, defined as proteinuria <0.3 g/day, serum Alb > 35g/L, and normal serum creatinine
Jatem-Escalante, 2021 [19]	Spain	Prospective	Patients diagnosed with nephrotic syndrome secondary to anti-PLA2R associated MN between 2010 and 2019	127 (44.9)	Not reported	6	A reduction of proteinuria ≥50% within the first 6 months after diagnosis

Abbreviations: Alb, Albumin; eGFR, estimated glomerular filtration rate; ESRD, end-stage renal disease; F/u, follow-up; MN, membranous nephropathy; PMN, primary membranous nephropathy.

**Table 2 jcm-12-00559-t002:** Performance assessment of prognostic prediction models for PMN.

Author/Year	Predictive Variables			Modeling Approaches	Performance		Validation	
	No.	Categorical Variables	Continuous Variables		Discrimination	Calibration	Internal Validation	External Validation
Zhang, 2020 [6]	4	hypertension, chronic tubulointerstitial injury	24 h-UP, serum uric acid	Cox regression analysis	AUC: 0.83, SPEC: 85.0%, SEN: 67.0%	Calibration plot	Bootstrap-corrected C-index: 0.84	Not reported
Hu, 2019 [7]	3		age, eGFR, proteinuria	Cox multivariate regression	AUC: 0.83	Not reported	Not reported	Not reported
Liu, 2022 [8]	4	age (≥ or <65 years)	proteinuria, sPLA2R-Ab levels, Uα1m/Cr	Logistic regression analysis	AUC: 0.89, SPEC: 72.2%, SEN: 95.2%, PPV: 44.4%, NPV: 98.5%, +LR: 3.43 (2.4–4.8), −LR: 0.066 (0.010–0.400)	Calibration plot	Bootstrap-corrected C-index: 0.869	Not reported
Pei, 1992 [5]	3	Persistent proteinuria	the slope of creatinine clearance, creatinine clearance	Logistic regression	SPEC: 88.0%, SEN: 66.0%, PPV: 66.0%, NPV: 88.0%	Not reported	Not reported	Finland cohort (ACC: 87%, SEN: 77%, SPEC: 89%, PPV: 59%, NPV: 95%); Italy cohort (ACC: 79%, SEN: 60%, SPEC: 92%, PPV: 64%, NPV: 82%) [20]
Branten, 2005 [9]	2	high Uβ2m (≥0.5μg/min), high UIgG (≥250 mg/24 h)		Cox regression	SPEC: 97.0%, SEN: 83.3%, PPV: 66.0%, NPV: 88.0%	Not reported	Not reported	Not reported
Tsai, 2019 [17]	2	Neutrophil-to-lymphocyte ratio (> or ≤3.34)		Cox proportional hazard regression	AUC: 0.68, SPEC: 79.7%, SEN: 55.8%	Not reported	Not reported	Not reported
Wu, 2022 [18]	6		sPLA2R-Ab, Creatinine, Urea, 24 h-UP, HDL-C, NC	Logistic regression	AUC: 0.91, SPEC: 88.0%, SEN: 81.5%	Not reported	Not reported	Not reported
Jatem-Escalante, 2021 [19]	2		baseline sPLA2R-Ab, sPLA2R-Ab reduction of ≥15% at 3 months	Logistic regression	AUC: 0.95, SPEC: 80.0%, SEN: 93.0%	Not reported	Not reported	Not reported

Abbreviations: ACC: accuracy; AUC, area under the ROC curve; eGFR, estimated glomerular filtration rate; HDL-C, high-density lipoprotein cholesterol; NC, neutrophil count; NPV, negative predictive value; PPV, positive predictive value; SEN, sensitivity; SPEC, specificity; sPLA2R-Ab, serum anti-phospholipase A_2_ receptor antibody; UIgG, urinary excretions of IgG; Uα1m/Cr, urinary α1-microglobulin corrected by creatinine; Uβ2m, urinary β2-microglobulin; 24 h-UP, 24 h-urine protein; +LR, positive likelihood ratio; −LR, negative likelihood ratio.

## Data Availability

The data analyzed or generated during the study is available from the corresponding author on reasonable request.

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
