# Peer review of "Prediction Models of Primary Membranous Nephropathy: A Systematic Review and Meta-Analysis"

_jcm, 2023, doi:10.3390/jcm12020559_

Round 1

Reviewer 1 Report

This is an interesting review and the authors have presented a systemic search and meta-analysis on Prediction models of primary membranous neuropathy.  The MEDLINE and EMBASE database have been extensively searched without any restriction of language or publication date. The authors have also focused on post treatment remission profile along with renal function deterioration; however, most of the literature reported prediction models are based on renal function deterioration. Also, the prediction models targeting disease emission may help with the patient selection for a treatment and customizing therapeutic procedures. The study has few limitations due to limited published prediction models on PMN as compared to other kidney diseases. High grade evidence from well-designed studies is required for PMN management. Over-all the review is impactful and well-written. The introduction is relevant, and theory based. Result and discussion sections are well written and explained. The study is registered on PROSPERO (identifier# CRD42022363539).

Reviewer 2 Report

The manuscript is worthwhile and need some minor revisions.

1-in table 1 in 2 and 3 studies and figures 3-4 , also in the text it is better to change renal function progression to renal function deterioration.

2- line 230 bad change to had

3- The discussion section is weak. It is better to improve this section.

Reviewer 3 Report

The authors have established a model to address the gap in the research and data available to study/understand PMN. The approach is clear and non-bias, they have addressed the shortcomings of their analysis and the manuscript will serve as a tool for others in the field to do such meta-analysis of the already available and accessible data from the community. The manuscript also addresses the deficiency of data and lack of research being done in the field of PMN which could serve as a call for researcher to focus more on PMN and collect data to fill this gap.
